# Next-Generation Leishmanization: Revisiting Molecular Targets for Selecting Genetically Engineered Live-Attenuated *Leishmania*

**DOI:** 10.3390/microorganisms11041043

**Published:** 2023-04-16

**Authors:** Paulo O. L. Moreira, Paula M. Nogueira, Rubens L. Monte-Neto

**Affiliations:** Biotechnology Applied to Pathogens (BAP), Instituto René Rachou, Fundação Oswaldo Cruz, Fiocruz Minas, Belo Horizonte 30190-009, Brazil; pmoreira@aluno.fiocruz.br (P.O.L.M.); paula.nogueira@fiocruz.br (P.M.N.)

**Keywords:** live-attenuated *Leishmania*, vaccine, leishmanization, genetic manipulation, CRISPR/Cas

## Abstract

Despite decades of research devoted to finding a vaccine against leishmaniasis, we are still lacking a safe and effective vaccine for humans. Given this scenario, the search for a new prophylaxis alternative for controlling leishmaniasis should be a global priority. Inspired by leishmanization—a first generation vaccine strategy where live *L. major* parasites are inoculated in the skin to protect against reinfection—live-attenuated *Leishmania* vaccine candidates are promising alternatives due to their robust elicited protective immune response. In addition, they do not cause disease and could provide long-term protection upon challenge with a virulent strain. The discovery of a precise and easy way to perform CRISPR/Cas-based gene editing allowed the selection of safer null mutant live-attenuated *Leishmania* parasites obtained by gene disruption. Here, we revisited molecular targets associated with the selection of live-attenuated vaccinal strains, discussing their function, their limiting factors and the ideal candidate for the next generation of genetically engineered live-attenuated *Leishmania* vaccines to control leishmaniasis.

## 1. Introduction

Present in 98 countries, threatening 350 million people at risk of infection, with 12 million infected people and 0.9 to 1.6 million new cases each year, leishmaniasis places among the global top ten neglected diseases, representing a serious worldwide public health problem [1]. It is noteworthy that these numbers could be under-reported, especially after the COVID-19 pandemic that affected the whole global primary health care system, interrupting the elimination program, and negatively impacting the diagnosis, treatment and control of the disease [2]. Transmitted by the bite of a female sand fly vector belonging to the Psychodidae family (*Phlebotomus* spp., *Lutzomyia* spp.), different species of *Leishmania* parasites can cause the diseases in humans, dogs and other mammals [3]. The spread of the disease and urban expansion is closely linked to poverty and environmental and climatological conditions. Leishmaniases are considered as a disease spectrum, including a wide range of clinical manifestations that can evolve to a fatal outcome depending on the *Leishmania* species involved and the integrity of the host immune response. Clinically, leishmaniases can be classified into different presentations, such as localized cutaneous leishmaniasis (CL), diffuse and disseminated CL, mucocutaneous leishmaniasis (MCL) and visceral leishmaniasis (VL) [4]. Canine leishmaniasis can be closely related to human cases in urban areas, since domestic dogs represent the most important reservoirs for the parasites [5].

In the Americas, the Mediterranean basin and the Middle East, VL due to *L. infantum* is considered primarily as a zoonosis, while it can be classified as an anthroponosis in India, other Asian countries and Africa, caused by *L. donovani*. Visceral leishmaniasis is the most fatal form if left untreated, especially in immunosuppressed patients. The symptoms include fever, weight loss, anemia and hepatosplenomegaly [6]. In contrast, CL can range from skin ulcers caused *by L. mexicana*, *L. panamensis*, *L. tropica* and *L. major*, to lesions that are difficult to treat, or more severe mucocutaneous lesions due to *L. braziliensis* or *L. guyanensis* and diffuse infections caused by *L. amazonensis* [4]. About 95% of CL cases occur in the Americas, the Mediterranean basin, the Middle East and central Asia [1].

Presenting a digenetic lifecycle, *Leishmania* parasites possess two main evolutive forms, varying morphologically and biochemically: promastigotes and amastigotes. The promastigotes are found within the invertebrate vector gut, and are elongated with emergent flagellum in the anterior portion that meets the mitochondrial kDNA at the flagellum basis in the cellular body. Promastigotes measure between 5 and 14 μm in length, and between 1.5 and 3.5 μm in width, and pass through morphological alterations during metacyclogenesis towards the most infective metacyclic promastigote forms [7,8]. After transmission via the bite of an infected female sand fly, the parasites are phagocyted by macrophages and other mononuclear phagocytes and turn into amastigotes due to changes in pH and temperature. In contrast, the clinically relevant amastigote forms present within the vertebrate host are rounded, measuring between 2 and 4 μm in diameter, with a short flagellum that barely emerges from the flagellar pocket and acts as an antenna for sensing environmental conditions [7,8,9,10,11].

Preventing and controlling the spread of leishmaniases is a difficult task. There are no current effective and safe vaccines available to prevent any clinical form of the disease in humans. In the absence of prophylaxis for humans, chemotherapy remains the main strategy for disease control. However, a few anti-*Leishmania* drugs are used in clinical practice, such as pentavalent antimony-derivatives (Sb^V^), amphotericin B, miltefosine and pentamidine [12]. Anti-*Leishmania* chemotherapy has several drawbacks, including serious side effects, the need for strict medical supervision, high costs and the emergence of drug-resistant parasites all over the world. Additionally, patients quitting treatment limits the success of chemotherapy even more [12,13,14].

Considering all the limitations associated with the clinically available anti-*Leishmania* arsenal, vaccination would be the main alternative for controlling and preventing the disease [15]. A central argument for the development of a vaccine against leishmaniasis is that patients who recover from the disease, either naturally or after drug treatment, develop immunity against the infection [16]. The strong immunity resulting from natural or drug-induced cures is commonly considered solid, although cases of reinfection have been documented, and justifies the effort on using live-attenuated parasites as a vaccine [17].

There are many promising studies devoted to finding a good vaccine alternative for leishmaniasis, based on different strategies, such as live-attenuated parasites, dead parasites, fractions of *Leishmania* antigens, recombinant proteins and DNA vaccines [18,19,20,21]. Despite this, we still lack a good, licensed, safe and highly effective marketed vaccine for leishmaniasis. Current knowledge is mainly based on animal models and cannot be easily extrapolated to humans [18]. Issues related to cost, antigenic complexity, genetic variability and the types of responses produced are limiting progress in a relevant direction [16,22]. In this sense, anti-*Leishmania* vaccines based on dead parasites, recombinant DNA, and protein vaccines fail to induce long-term protection [23]. It is a consensus that parasite persistence is important for sustaining a protective, durable, anti-*Leishmania* immune response [23,24,25]. This has inspired researchers to revisit leishmanization—a vaccination method widely used in the Middle East in the past, which consists of inoculating infective *L. major* in the skin, leading to a lesion and self-healing process while protecting against reinfection. However, the modern proposal is to use live, genetically modified *Leishmania* parasites as a vaccine strategy [26,27,28,29]. Protection is mediated by the host’s immune system, capable of generating memory cells against subsequent infection via the bite of the sand fly vector containing the infective forms of the parasite [30].

Indeed, several studies using live-attenuated *Leishmania* have shown it to be a good strategy for promoting a long-lasting protective immune response, meeting the requirements to be used as vaccine [27,31,32,33,34,35,36,37]. Infection with live-attenuated *Leishmania* is similar to infection with virulent parasites, but with the advantages of preventing disease while allowing the host immune system to interact with a wide range of *Leishmania* antigens in the development of protective immunity [28]. Since 2012, with the emergence of CRISPR/Cas gene editing technology [38], and more recently with the democratization of CRISPR-based techniques applied to *Leishmania* parasites [39,40], it is now possible to select safer attenuated parasites more precisely [30,34]. Modern leishmanization using genetically engineered live-attenuated parasites could be used as a promising vaccine for both humans and dogs. The mechanisms behind parasite persistence, even from an attenuated mutant, together with long-lasting protection, is yet to be described. However, promastigote to amastigote differentiation within the host cell seems to be important for mounting the protective phenotype [33,35].

Here, we revisit the gene targets involved in the selection of live-attenuated *Leishmania* parasites, providing the state of the art in the field, and highlighting promising vaccine candidates. We also discuss gene function and its role in the parasite’s biology, together with a functional analysis of the 54 selected candidates. Only 10% of live-attenuated *Leishmania* null knockout mutants (5/54) were filtered as presenting both protective immune response (in vitro and/or in vivo) and in vivo protection upon reinfection. We highlight the limiting factors for developing a live-attenuated *Leishmania* vaccine strain and suggest the desired features in the target product profile for the next generation of leishmanization.

## 2. Immune Response for Controlling Leishmaniases

Several factors influence the resistance or susceptibility to *Leishmania* infection, such as parasite species and strains; genetic variations; host immunological status; inoculum size; the location and number of vector bites; the associated vector microbiota; saliva proteins; and extracellular vesicles [41,42,43,44]. Additionally, parasites can evade the host immune system, aiding in the establishment of infection [45,46]. T helper lymphocytes play a critical role in orchestrating the immune response against the parasite in mammalian hosts. Their polarization towards either Th1 or Th2 response has been shown to be a major determinant of disease outcome, mainly based on investigations using *L. major* [47,48].

The innate immune response of the host determines the resistance or susceptibility to infection, mediated by macrophages, neutrophils, NK cells and dendritic cells (DC). The production of IFN-γ, TNF-α, IL-1b, IL-6, IL-12, IL-23 and nitric oxide (NO) are essential for killing the parasite inside infected macrophages (Th1 response) and provides immunity to reinfection. In contrast, the production of some key cytokines, including TGF-β, IL-4, IL-10 and IL-13, is correlated with an anti-inflammatory phenotype and may promote an alternative macrophage activation, which favors parasite persistence (Th2 response) [49]. According to a study by Kane and Mosser (2001), the increased expression of IL-10 during *Leishmania* infection is associated with the severity and progression of the disease [50]. Therefore pro-inflammatory Th1 immune response, together with modulated Th2, with an increased Th1/Th2 ratio, is essential for controlling leishmaniasis [48]. Humoral response with seroconversion on anti-*Leishmania*-specific IgG and its subclasses is produced in low levels during CL but is positively modulated in VL cases, although its role is not completely understood [49]. Therefore, the ability to elicit a cell-mediated immune response, including effector and central memory cells, is essential for the development of an anti-*Leishmania* vaccine.

As people clinically cured of leishmaniasis develop lifelong protection, the development of a vaccine using live-attenuated parasites through leishmanization is an attractive approach. Attenuated parasites closely mimic infection with virulent parasites, which can trigger a potent activation of the immune system and are effective against different species of *Leishmania*. An example of this efficacy was demonstrated when using *L. tarentolae*, a non-pathogenic form for humans, as a vaccine. By inoculating BALB/c mouse models, the vaccine was able to efficiently target dendritic cells and lymphoid organs, thus increasing antigen presentation and influencing the magnitude and quality of the T-cell-mediated immune response [51]. However, there are concerns when using live-attenuated vaccines, such as random genetic mutations and potential reversion to virulence, as well as the persistence of an asymptomatic infection, especially in immunocompromised individuals. This can make diagnosis difficult and may increase the risk of clinical reversion [51], with protection and safety as the main concerns. In this sense, using growth-arrested genetically modified parasites is promising as mutant parasites generally do not revert to virulence in animal models, even under conditions of induced immunosuppression, due to the complete deletion of essential gene(s) [52].

Earlier studies demonstrated that live-attenuated *Leishmania* mutants had various Th1 stimulatory proteins upregulated by vaccination using either Centrin1-, HSP70- or p27-deleted parasites. These strains exhibit a significant immune response activity, such as displaying an enhanced expression of INF-γ, TNF-a, elicited NK, CD4 and CD8, and induce the production of IL-6 and IL-12 [33,53,54,55]. Likewise, live-attenuated parasites induce significantly lower levels of Th2 cytokines such as IL-10 and IL-4 compared to virulent parasites [29]. Furthermore, although antibodies do not play a specific protective role in *Leishmaniasis*, an increase in IgG2a/IgG1 has been associated with protective immunity upon vaccination with live-attenuated parasites followed by challenge with a virulent strain [29,56,57]. These responses are able to confer long-term protection in some animal models against the infection with virulent *Leishmania* parasites.

Many studies have identified promising live-attenuated mutant strains that are able to survive as promastigote, can infect host cells and become amastigotes and elicit an immune response but fail to maintain infection. Growth-arrested attenuated parasites should not cause any disease and would ideally reduce parasitemia, maintaining a protective immune response with increased Th1 cytokine production by CD4+ and CD8+ cells and an increased IFN-γ/IL-10 ratio. One good advantage of this is the potential to protect against reinfection with different species of virulent *Leishmania* strains, acting as a pan-*Leishmania* vaccine [29]. In summary, an effective live-attenuated vaccine requires a parasite with growth and infectivity similar to a wild-type parasite in the initial steps of infection, capable of presenting the range of antigens to the vertebrate host that are necessary to mount a long-term, protective immune response.

## 3. Genetic Manipulation for Selecting Live-Attenuated *Leishmania*

The ability to manipulate the genome of *Leishmania* parasites for selecting genetically modified mutants by disrupting virulence-factor-encoding genes sheds light on modern leishmanization. This opens new avenues for live-attenuated parasites as vaccine candidates, which is a powerful tool for the development of next-generation vaccines against leishmaniasis, inspired by old strategies. Independent functional-genomics-based studies found important targets by serendipity, and these were further applied as potential vaccine candidates [58,59,60].

The first report on gene deletion in *Leishmania* was published in 1990. This involved homologous recombination, using linear fragments of DNA containing a gene that confers resistance to neomycin (resistance marker) together with the untranslated regions (5′ and 3′UTRs) that flank the gene target dihydrofolate reductase thymidylate synthase (DHFR-TS), allowing gene replacement mediated by the recombination process. This was possible due to the ability of the parasite to efficiently replace homologous genes efficiently with exogenous DNA [61]. Several studies have reported the use of targeted gene deletion strategies for the development of attenuated vaccine strains for *Leishmania*. However, the genetic manipulation of trypanosomatids has historically been challenging, due to factors such as the presence of supernumerary chromosomes (polyploidy), the incredible genome plasticity and the emergence of compensatory mutations that could contribute to the failure of conventional gene replacement techniques, especially for essential genes [62,63,64,65,66,67]. The conventional deletion strategy using homologous recombination for allelic replacement involves multiple cloning steps, long flanking regions for homology-directed repair (HDR) and at least two rounds of transfections with different resistance markers for the selection of null mutants [68]. Despite these limitations, and depending on the target, traditional homologous recombination is still used due to its cost effectiveness and standardization established over the years, showing great efficiency in deleting genes.

The recently reported CRISPR/Cas gene-editing-based system has revolutionized the way we manipulate genomes, including in *Leishmania* parasites. This robust and precise technique allows gene disruption or tagging by knocking in (or out) multiple alleles at once with one single nucleotide precision [38]. This prompted researchers to reselect live-attenuated *Leishmania* strains more quickly, with improved precision and safety, including the selection of markerless free mutants [29]. The recent adaptation of CRISPR technology to different protist models has played a fundamental role in the functional study of their proteins, metabolic pathways and the understanding of their biology, and this will facilitate the search for new chemotherapeutic and therapeutic targets [69]. The technique allows for rapid protein labeling and gene knockout in several species, without the need for DNA cloning, and with only one round of transfection to delete all alleles [40]. Recent advances based on the CRISPR/Cas system have allowed for genome-wide loss-of-function screening without the need for DNA double-strand break, homologous recombination or donor DNA, by using LeishBASEedit (http://www.leishgedit.net/LeishBASEedit/Home.html, accessed on 17 March 2023)—a cytosine base editor toolbox that targets single or multicopy *Leishmania* genes, resulting in null mutant parasites from one sgRNA with an efficiency rate of 100% [39].

The first successful use of CRISPR for editing a *Leishmania* genome dates back to 2015, when Sollelis and colleagues were able to efficiently delete one of the paraflagellar rod homologues (PFR)2 in *L. major* using a single round of transfection [70]. Following this discovery, more recent studies on attenuation have utilized CRISPR to generate null mutants with high efficiency. Among the deleted targets are Centrin1 [29,34,36,71], eukaryotic translation initiation factor 4E-1 (EIF4E1) [72] and the noncatalytic component of the GPI-mannosyltransferase (GPI-MTI) complex [73].

A good hint for selecting gene candidates for disruption towards the development of live-attenuated parasites is to choose essential targets specific to the amastigote stage, without disturbing promastigote forms. Frequently, we see reports concluding gene essentiality in promastigote forms of *Leishmania* simply based on the fact that the parasites did not survive after one attempt at disrupting a given gene using the CRISPR/Cas system. It is important to mention that these tools do not work in the same way for different cell types and targets. Unlike other organisms that benefit from non-homologous end joining (NHEJ) as a DNA repair mechanism during CRISPR/Cas-mediated gene editing, *Leishmania* parasites rely on microhomology-mediated end joining (MMEJ), single-strand annealing (SSA) or homology-directed repair (HDR)—in this latter case, when in presence of a DNA donor cassette to replace the deleted region—upon double-strand DNA breaks due to Cas9 nuclease [74,75]. Depending on the strategy adopted and on the selected gene target, it can interfere with the success of the technique. An initial failure in CRISPR/Cas gene editing does not confirm that a gene is essential; for that we recommend the use of an episomal rescue vector strategy in order to validate it [76,77].

Moreover, gene deletion is also useful in functional genomic approaches to investigating molecular functions linked to metabolism, signaling, cell division, differentiation, etc. Understanding the protein functions, especially from unknown targets by deleting their encoding gene, leads to a better understanding of parasite biology, which can evolve to the identification of essential factors supporting the identification of new therapeutic targets, or can be used for developing attenuated vaccine strains for controlling leishmaniasis. It is worth noting that deleting these targets for vaccine studies requires the insertion of donor DNA containing a selection marker that confers resistance to pre-established antibiotics. This is necessary in order to replace the gene of interest and select the attenuated mutant. However, the presence of antibiotic resistance genes in any attenuated vaccine is not acceptable for human testing by regulatory agencies. Considering this issue, a recent study by Zhang and colleagues (2020) succeeded in obtaining an attenuated strain of *L. major* without the presence of selection markers and off-target effects. This knockout of *Lm*Cen^−/−^ provides protection against virulent strains naturally infected via sand fly bites without causing harm to the vertebrate host, demonstrating its safety and efficacy for future human trials [34].

These arguments demonstrate that it is possible to be inspired by first-generation vaccine strategies that provide robust immune responses, such as leishmanization, which can be combined with modern genetic manipulation techniques to design, study and advance vaccine development, especially when dealing with neglected diseases that affect millions of people worldwide.

## 4. Genetically Engineered Live-Attenuated Vaccine Candidates

Vaccines composed of attenuated viruses and bacteria are considered the gold standard against intracellular pathogens [78]. Several attenuated parasites have also been tested in animal models. Radio-attenuated and biochemically altered parasites have been shown to provide good protection in mice and hamsters, even in the absence of adjuvants [79]. However, concerns about the possibility of re-establishing virulence pose major ethical hurdles for their use in humans.

Targeted genetic modification of specific virulence-associated genes for selecting live-attenuated *Leishmania* parasites is a potent strategy for improving safety profiles. Here, the parasite should induce a strong and protective memory response, but must be eliminated and must not cause disease, remaining for long enough to mount an immune response [28]. In contrast, parasite persistence upon vaccination with live-attenuated strains seems to be common in the studied models, although the mechanisms behind this persistence have not been fully described. Despite this phenomenon, no return of virulence has been described in experimental modern leishmanization. There is no prerequisite for choosing the target gene to be deleted in order to select an attenuated strain that is capable of infecting host cells and generating a protective immune response. Protein target examples vary from surface molecules to metabolic pathway enzymes, cell signaling proteins, carrier proteins, cytoskeleton-associated, etc. The important thing is that the targeted disrupted gene leads to the effective selection of attenuated parasites that can infect but do not cause any clinical manifestation. In this regard, live-attenuated vaccine candidates should be tested in different experimental models in order to obtain a better understanding of their potential to protect against not only one disease, but a spectrum of them, which is the case of leishmaniases.

Many gene targets have been identified as promising for selecting live-attenuated parasites in different *Leishmania* species. Table 1 summarizes 54 selected gene targets in the context of live-attenuated parasites from reports between 1995 and the present. It compiles the function or molecular pathway that the product is associated with, lists the *Leishmania* species involved, together with the experimental model used in each study and the obtained phenotype upon infection or vaccination. To facilitate data interpretation from Table 1, we created Figure 1, which illustrates the models used to study live-attenuated *Leishmania* and highlights the selection of a few promising candidates. Additionally, we performed a functional gene ontology (GO)-based analysis of all gene targets. This is presented in Appendix A, using *L. major* Friedlin and *L. donovani* BPK282A1 strains as references, since the majority of studies (70%) were carried out using these two Old World species (Figure 1). A GO enrichment analysis, including the terms Biological Process, Molecular Function and Cellular Compartment, was performed using TriTrypDB (www.tritrypdb.org, accessed on 15 March 2023), considering curated and computed evidence with a *p*-value cut-off of 0.05. Metabolic pathway enrichment was also performed with the same tool using KEEG and MetaCyc as pathway sources. The resulting word cloud, hierarchically based on *p*-value, is shown for each GO term or metabolic-associated pathway (Appendix A).

The following target description is based on the candidates presented in Table 1.

Several gene targets that encode proteins important for the selection of attenuated parasites have been reported, including arabino1,4-lactone oxidase (ALO) protein [32,83], cytochrome c oxidase complex component p27 [84,85,86], HSP70 type II heat shock protein [31,56,102,103,104], centrin1 [29,33,34,35,36,53,54,71,117,118,119] and kharon1 [57,58,59]. Other target candidates have also shown potential, such as biopterin transporter (BT1) [37], dihydrofolate reductase/thymidylate synthase (DHFR-TS) [88] and Golgi GDP-mannose transporter (LPG2) [92,137]. However, unexpected gains of virulence, remaining alleles or the failure to protect non-rodent experimental models discouraged further investigations [28]. Some other recent candidates, such as the DDX3 DEAD-box RNA helicase (Hel67) [27], despite having some promising results in hamsters, require deeper understanding in relation to immune response and prophylaxis. Indeed, most candidates are still in the early stages of development and require further study to determine their efficacy in protecting against *Leishmania* infection.

An important point to consider is gene essentiality, which can lead to debates about whether a gene is necessary in all life stages of the parasite, or only in specific growth conditions [138]. For instance, the family proteins (SIR2) that perform a unique NAD-dependent histone deacetylase activity are crucial for the parasite’s survival, and the knockout of both alleles in *L. infantum* is not possible as they do not survive as promastigotes. However, the knockout of a single allele (*Li*SIR2^+/−^) drastically affects the proliferation of axenic amastigotes and their ability to survive within macrophages, and also offers protection to vaccinated mice [77,108].

The mitochondrial carrier protein (MIT1), which is involved in mitochondrial iron uptake, an essential cofactor of respiratory chain proteins, is critical for the function and maintenance of redox balance. It has been shown to be essential for promastigotes of *L. amazonensis*, where only a mutant lacking an allele (LMIT1/Δ*lmit1*) can be selected. This mutant was unable to survive as intracellular amastigotes within macrophages or cause injury as it is susceptible to ROS toxicity [112].

Type I peptidase (SPaseI) is an endopeptidase responsible for removing the signal peptide from secretory pre-proteins, releasing the mature proteins into the intra- or extracellular space. Its role is essential for the survival of the parasite. In *L. major*, it has been demonstrated that it is not possible to delete both alleles of the SPaseI gene. Only a heterozygous single knockout mutant (*Lmj*^+/−^) could be obtained. However, these mutants significantly reduced the levels of infectivity in macrophages and did not cause lesions in mice [134].

Cysteine peptidases (CPs) play important roles in *L. mexicana* virulence, where the knockout of its natural inhibitor (CPI) increased infectivity and the generation of marked, non-healing lesions in mice [139]. However, overexpression of ICP generated attenuated *L. mexicana* species, with increased Th1 response in mice [135].

The *L. major Lm*DNA16 locus encodes a family of hydrophilic surface proteins that are related to the HASP genes (HASPA1, HASPA2 and HASPB) as well as endoplasmic reticulum and mitochondrial outer membrane proteins that are linked to SHERP1 and SHERP2 genes. Deletion of the *Lm*DNA16 locus showed that the mutants are as virulent in macrophages as the wild-type parasites, and are capable of forming lesions in mice. However, a decreased expression of the surface glycoprotein GP63 was observed upon *Lmi*DNA16 locus overexpression, associated with reduced parasitemia in mice [136].

Although these targets elicit an immune response in the host, reduce parasitemia and do not cause lesions in murine models, they are not considered attractive candidates because the safety of such mutants cannot be guaranteed. These mutants still carry the unique alleles of the wild-type gene, which can revert to the wild-type genotype and regain virulence. In this sense, it is essential to develop parasites that are attenuated by a complete disruption of both gene target alleles, reducing the chances of causing disease by reverting to virulence.

As previously discussed, amastigote-specific gene essentiality can be explored to screen live-attenuated parasites. Features of potential gene targets based on their molecular function and their role in *Leishmania* biology must be considered in order to increase the chances of validating a good candidate. In this sense, we revisited some targets, pinpointing biochemical features that favor the selection of vaccine strains.

In this context, Centrin1 (Cen1) knockout *Leishmania* is by far the most studied live-attenuated vaccine candidate that has already reached clinical trials [140]. The calcium-binding basal body-associated protein in *L. donovani*, Centrin1, is involved in cell division via centrosome duplication and segregation [121]. Deletion of Cen1 in *L. donovani* (*Ld*Cen1^−/−^) does not affect promastigotes in vitro. However, the growth of the amastigotes is blocked, and axenic amastigotes showed cell cycle arrest in the G2/M phase, indicating a failure in basal body duplication and cytokinesis, resulting in multinucleated cells. This may trigger the programmed cell death pathway of the parasite [122]. A very good and complete review of Cen1-KO mutant history was published recently and is well worth reading [29].

The enzyme Arabino-1, 4-lactone oxidase (ALO) is known to catalyze the terminal step of the ascorbate biosynthesis pathway [83], which is an important antioxidant that can directly metabolize reactive oxygen species (ROS) and mediate electron transfer to ascorbate-dependent peroxidases [141,142]. In addition, ascorbate plays a key role in controlling the parasite’s differentiation and survival in host macrophages, since the knockout of the gene that encodes for the ALO protein in *L. donovani* showed a reduction in macrophage infection, absence of ALO activity and reduced ascorbate levels [83]. After infection, pro-inflammatory cytokines, such as tumor necrosis factor-α (TNF-α) and interferon-γ activate macrophages, produce ROS and reactive nitrogen species (RNS) regulated by the expression of nitric oxide synthase (NO) [143]. This suggests that ascorbate, which acts as an electron donor for these dependent peroxidases and the enzyme that catalyzes their synthesis (ALO), may play a decisive role in the process of establishing the parasite [140].

Cytochrome c oxidase complex component 27 (p27) is an inner mitochondrial membrane protein with a more abundant expression in metacyclic promastigote forms of *Leishmania*. The protein is an important component of an active oxidase complex and is involved in ATP synthesis specifically in these forms. Deficient mutants of *L. donovani* (*Ld*p27^−/−^) have been shown to be significantly impaired in these activities due to failure in the electron transport chain and, consequently, in cellular respiration [84,144].

Another enzyme that contributes to attenuation upon its gene disruption is fructose-1,6-bisphosphatase (FBP), which catalyzes the final committed step of the gluconeogenic pathway, converting fructose-1,6-bisphosphate into fructose-6-phosphate. This can continue for carbohydrate synthesis and/or for the synthesis of inositolphospholipids and surface glycoconjugates [98]. FBP in *L. major* and *L. donovani* is constitutively expressed in both extracellular and intracellular stages. However, null mutants (*Lmj*Δfbp and *Ld*Δfbp) of FBP in promastigote forms were unable to survive in the absence of hexose, and amastigotes were unable to multiply in the macrophage phagolysosome. This suggests that *Leishmania* amastigotes reside in a glucose-poor phagosome and depend on non-glycosylated carbon sources for their survival [97,98].

The Kharon (KH) protein is responsible for targeting the *Lmx*GT1 glucose transporter from the flagellar pocket to the flagellum membrane in *L. mexicana* through interaction with the proximal portion of the flagellar axoneme and recognition of the flagellar-targeting domain of *Lmx*GT1. The absence of KH in *L. mexicana Lm*ΔKh1 showed the accumulation of this permease in the flagellar pocket [58]. Although *Lm*ΔKh1 and *L. infantum Li*ΔKh1 mutants were able to survive as promastigotes and infect macrophages, they failed during cytokinesis as intra-macrophagic amastigotes, resulting in the formation of several multinucleated cells with retention in the G2/M phase, and they did not survive within the host cell in vitro [57,59]. Due to the different subcellular localizations of KH and evidence of alternative functions in different trypanosomatids, it is postulated that KH is characterized as a multitask protein exerting cytoskeleton-associated functions [59,145].

It is important to mention that the deletion of both alleles of a given gene does not necessarily mean that it will exhibit attenuation features. Sometimes the reverse effect may occur, as demonstrated by the deletion of the enzyme ascorbate peroxidase in *L. major* (*Lmj*APX), which plays a central role in the redox defense of *Leishmania*. APX knockouts showed an increase in metacyclic promastigotes in culture. After infection, they exhibited hypervirulence in macrophages and during the inoculation of BALB/c mice [146]. Deletion of the gene encoding an ATP-binding cassette transporter protein subfamily C in *L. donovani* resulted in an increase in the growth of axenic amastigotes and increased virulence in infected mice [147]. Arginine methylation is a conserved post-translational modification carried out by protein arginine methyltransferases (PRMTs). Among the five coding homologues of PRMT in *L. major*, LmjPRMT7 is a cytosolic protein that associates with several RNA-binding proteins. Its deletion increases virulence and, consequently, increases lesions in mice compared to wild-type strains [148].

These studies demonstrate the importance of understanding the function and pathways of each target to develop effective attenuated vaccines. Parasites often have compensatory strategies that enable their survival in the vertebrate host, resulting in disease manifestations similar to those of wild-type lineages.

## 5. Limiting Factors for Selecting Live-Attenuated *Leishmania* Vaccine Strains

Several immunological aspects of leishmaniasis have been explored using experimental animal models, such as mice, hamsters, dogs and non-human primates. The correct pipeline for vaccine development must be driven by the choice of the experimental model based on the answers it can give, considering the advantages and drawbacks associated with each particular model (e.g., the costs of maintaining mice, hamsters or dogs and performing experimental infections using infected sand flies instead of needles). A full understanding of the benefits and drawbacks of different strategies is crucial for choosing the proper match. The outcome of experimental infection is based not only on *Leishmania* virulence but also on a combination of factors that include host species and immune status, the inoculum route, inoculum size, the organ chosen, efficacy evaluation and the time after infection [149,150]. In vitro and ex vivo testing can be faster and cheaper than in vivo, but may not fully reflect the complexity of the immune response to a vaccine in a living organism and may not consider the context of other cells or tissues in the in vivo models.

The initial tests for the experimental vaccine are mostly performed in murine models due to their advantages regarding rapid reproduction, ease of maintenance and manipulation and a large collection of genetically identical inbred strains. These advantages corroborate the reproducibility of the studies [151]. Although experimental murine models for VL and CL have been established, they do not accurately mimic human disease. In comparison to hamsters and dogs, mice typically exhibit fewer minor clinical signs and symptoms during *Leishmania* infection, depending on the inoculum size [152]. Experimental infection in golden hamsters is considered the best animal model for canine and human VL, with similar clinic-pathological features and inoculum routes [28,149,153]. However, the main negative point is the lack of immunological reagents for hamsters (antibodies, cell markers and cytokines), making it difficult to evaluate the cellular immune response and the cytokine production induced after vaccination [154]. To overcome this issue, cross reactivity between anti-mouse mAbs and hamster molecules can be considered for characterizing immune response in the latter [155]. Another solution is the use of an RT-qPCR technique addressed to mRNA that codes for immune response targets in hamsters [156].

Given that domestic dogs are important reservoirs for VL, the use of them as an experimental model in subsequent tests to study anti-*Leishmania* vaccines has led to the elucidation of the role of the immune system, which is considered a promising strategy for the control of both the human and canine disease [157,158]. Remarkably, there are few reports considering dogs as models to study live-attenuated *Leishmania* (Figure 1). Recent research using *Ld*Cen^−/−^ was important as it gave us a better understanding of immune response against VL, which showed higher antibody titers, the induction of T and B cell proliferation and type 1 polarization. Immunization with a single dose of *Ld*Cen^−/−^ reduced parasite burden in dogs, suggesting a control of parasite replication and a prevention of severe disease upon challenge with virulent strains even in the long term [53,117]. In a recent work, the protection elicited by live-attenuated *Lm*p27^−/−^ remained at the follow up 12-months after the vaccination in dogs, with no clinical signs of disease and increased protective response [87]. In contrast, the limitations of using dogs as models can raise ethical concerns; they are expensive and require housing and care. Additionally, immunosuppressed dogs may be more susceptible to vaccination side effects, which affects parasite clearance, resulting in positive xenodiagnostics [159,160].

Some evidence suggests that live-attenuated parasites of *Leishmania* species associated with VL can provide heterologous protection against CL-causing species in different animal models [35]. These findings were confirmed when *Ld*Cen^−/−^ vaccination conferred cross protection against infection with a heterologous challenge with *L. braziliensis*, *L. mexicana* and *L. infantum* [54,161].

The use of non-human primate models for the evaluation of potential vaccines is often used prior to human clinical trials. The phylogenetic closeness to humans makes them attractive models for assessing the pathogenicity of leishmaniasis [90,162]. Amaral and colleagues (2002) evaluated the efficacy of live-attenuated DHFR-TS-deficient *L. major* in protecting rhesus macaques against reinfection with a virulent *L. major*. In this report, the vaccinated rhesus group had low antibody levels, presented skin lesions and was not protected upon challenge with a virulent strain, although the vaccine protected mice against CL [90]. The use of non-human primates in scientific research raises ethical concerns that must be carefully considered. Additionally, they are more expensive and require proper infrastructure [163]. For these reasons, the use of non-human primates in research is highly regulated and usually limited. In contrast, murine, hamster and dog models are typically less restricted when compared to non-human primates.

To evaluate the impact of parasitic burden in experimentally vaccinated animals, the main parameter for assessing efficacy is the quantification of parasite load. This can be achieved by using real-time quantitative PCR to measure the amount of parasite DNA in a tissue sample, which is then multiplied by the total weight of the organ. This technique can be used to diagnose or monitor the evolution of infection in mice, dogs and humans with high sensitivity, accuracy and reproducibility [149]. While molecular approaches to quantifying parasite load in vaccinated animals are extremely relevant, it is also important to determine whether the genetic material detected corresponds to live or dead parasites. Since the limited persistence of live-attenuated parasites is an important factor for the development of durable protective immunity, studies have limitations in determining long-term parasite persistence. *Ld*Cen^−/−^ parasites have been shown to persist for a maximum of 5–8 weeks in mice [33]. The parasite load can also be assessed by limiting dilution assay (LDA) for detecting viable parasites; however, this technique can be limited by cross contamination and low sensitivity when few parasites are present in the sample. Depending on the approach, it needs to be invasive or applied to euthanized animals. The LDA technique is commonly less sensitive when compared with quantitative PCR [149].

Another important issue concerning experimental infection with live-attenuated *Leishmania* is whether the infection can be reactivated, particularly in immunodeficient individuals. The selection of null mutant parasites is fundamental for avoiding this phenotype.

Most vaccination and challenge assays are performed using needles to inject a large number of parasites by different routes (intraperitoneal, intravenous, intradermal, intramuscular or subcutaneous). In general, the intradermal route of administration can offer improved protective immunity and be more efficient with smaller amounts of vaccine [164]. Although needles are commonly used to administer live-attenuated vaccines, to mimic the natural conditions of a reinfection we ideally need to include infected sand flies when performing an experimental challenge. This is very difficult to achieve as it is not easy to acquire all the necessary facilities. There are few locations in the world that perform natural infection, and this reinforces the global challenge.

Before we consider testing a live-attenuated *Leishmania* vaccine to be tested in humans, we must ensure that the vaccine candidate is safe; that it presents genetic stability without any chances of reverting virulence; that it cannot be transmitted to other potential hosts; that it can be preserved where it is most needed; and that it ensures stability and presents limited parasite persistence, long enough to provide the desired protective immunity. Compared with the ancient leishmanization practice, modern leishmanization is safer—since attenuation is acquired by precise gene editing—does not cause lesions or any disease sign and promotes pan-protection for leishmaniasis caused by different species (heterologous protection). Leishmanization induces parasite-independent memory skin-resident T cells, which confer distinct protection based on their capacity to respond almost immediately upon challenge [30]. Modern leishmanization is still limited by the parasite’s turn-off control and the parasite’s persistence.

Overall, the establishment of guidelines for the development of live-attenuated *Leishmania* has the potential to significantly improve the timeline of candidates already in the pipeline, by providing a clear route to safety and efficacy studies before conducting clinical trials in humans.

## 6. Target Product Profile for Live-Attenuated *Leishmania* Vaccines

An ideal human vaccine based on live-attenuated *Leishmania* must

(I)Be a null mutant *Leishmania* parasite in which full gene allele disruption is achieved without any antibiotic resistance marker during parasite selection [34];(II)Elicit protective immunity and prevent disease development upon reinfection with a virulent strain;(III)Be a pan-*Leishmania* vaccine that protects against homologous and heterologous challenges or, at least, protects against the relevant circulating species;(IV)Be compatible with good manufacturing practices (GMP) in order to comply with regulatory requirements [35];(V)Ensure parasite persistence long enough to maintain protective immunity in safe parasite load levels to avoid any transmission;(VI)Be administered as a single dose that maintains long-term protection;(VII)Not cause lesions or any signs of disease;(VIII)Show no side effects that require health care monitoring;(IX)Present no contraindications;(X)Be stable under climate conditions of the regions it is needed most.

## 7. Concluding Remarks

The selection of live-attenuated *Leishmania* parasites for use in leishmaniasis prophylaxis is an old strategy and is inspired by the practice of leishmanization, which has gained special attention recently due to the refinement of gene editing technologies, with an emphasis on CRISPR/Cas-based systems. Upon revisiting more than 50 selected gene targets whose editing leads to live-attenuated parasites, we can now document the state of the art techniques in the field, identifying patterns and important gaps and highlighting actions that are needed to develop live-attenuated vaccines against leishmaniases. Among the studies, the vast majority were performed using *L. donovani* and *L. major* parasites, with very few on New World species such as *L. amazonensis* and *L. braziliensis.* This could be negligible in practical terms, since live-attenuated *L. donovani* mutants could protect against heterologous challenges using New World species. However, the genetic manipulation of *L.* (*Viannia*) *braziliensis* in order to obtain live-attenuated vaccine strains could be of special interest as the only representant of the *Viannia* subgenus and could provide mechanistic insights, since it is a very unique species presenting natural hybrids and RNAi functional machinery and is associated with CL and MCL in the Americas. Almost all in vivo efficacy studies were based on BALB/c mice, which probably reflects the early stages of development. Few studies have pushed forward the use of non-rodent models, and we need to advance on this in order to mimic human leishmaniasis, as it is more realistic and supports translational science. Although studies identified the potential of live-attenuated vaccine candidates for leishmaniasis, several studies concentrated on the proof of concept, and only a few were dedicated to a deep understanding of the protective immune response. In this sense, experimental murine models of CL are preferred since they are easy to perform and provide evidence of protective phenotypes without the need for expensive reagents or equipment. However, we should expend more energy on applying the vaccine candidates to initial protection against VL challenge, as this is the most fatal form of the disease and contributes to the majority of deaths by leishmaniases worldwide. A functional analysis of the gene candidates presented in Table 1 was performed and revealed purine metabolism as one of the main enriched pathways among them. Indeed, *Leishmania* are purine auxotroph organisms, which highlights the importance of these pathways for developing antileishmanial drugs and vaccine candidates. However, by filtering the candidates by efficacy (increased protective immune response and in vivo protection upon challenge with virulent *Leishmania*), we found Cen1 to be one of the most promising, together with ALO, P27 and FBP. Indeed, Cen1 is the most advanced candidate for the first modern live-attenuated *Leishmania* vaccine supported by several studies, but we urgently need to advance other promising targets and add new potential candidates to the pipeline.

## Figures and Tables

**Figure 1 microorganisms-11-01043-f001:**
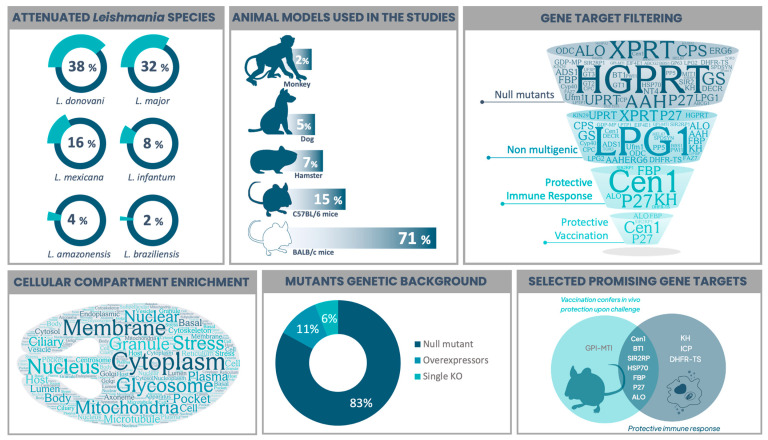
Compiled features from studies on the selection of live-attenuated *Leishmania* parasites. The viscerotropic species *L. donovavi* and the dermotropic *L. major* were the main studied parasites, followed by New World representant species. Most in vivo assays were performed in mice, with BALB/c being the model used in more than 70% of the studies. Gene targets whose disruption leads to an attenuated phenotype were filtered according to (I) the selection of null (full gene knockouts) mutants (45/54); (II) the presence of only non-multigenic targets (35/45); (III) the experimental evidence of eliciting protective immune response (in vitro and/or in vivo) (7/35); and (IV) the evidence of protection when using the live-attenuated *Leishmania* as a vaccine strategy upon challenge with a virulent strain (5/7). Cellular compartment enrichment analysis was performed based on all gene targets and is presented as a word cloud to show what they are most associated with. The majority of *Leishmania* parasites were selected as null mutants, where both alleles are disrupted. Putative essential targets did not result in full gene disruption, but the single knockout version (3 studies) was enough to achieve attenuation. By contrast, gene overexpression led to attenuated phenotypes in two studies. Combining the two main desired features, “evidence of anti-*Leishmania* protective immune response” with “in vivo protection upon challenge with a virulent strain”, we highlighted promising vaccine target candidates. Figure 1 was prepared based on the data presented in Table 1, which summarizes reports from 1995 to 2022. Word clouds were produced using wordart.com. The Venn diagram was prepared using freetools.textmagic.com. Some icons were obtained from flaticon.com.

**Table 1 microorganisms-11-01043-t001:** Genetically modified *Leishmania* parasites where gene editing leads to attenuation. The function or metabolic pathway associated with each gene target is presented together with the *Leishmania* species, experimental model used and the resulting phenotype (in vitro and in vivo).

Gene ID	Product	Function	Organism	Model	Phenotype	Ref.
*Targets whose disruption leads to attenuation profile*						
LdBPK_210980.1 LdBPK_210990.1 LdBPK_352200.1	Hypoxanthine-guanine phosphorybosyl-transferase (**HGPRT**) Xantine phosphoribosyltransferase (**XPRT**) Adenine Aminohydrolase (**AAH**)	De novo purine synthesis	*L. donovani*	Murine BMDM BALB/c mice	Unable to sustain infection in vitro ↓ parasitemia	[80,81]
LdBPK_160590.1 LdBPK_341110.1	Carbamoyl phosphate synthetase (**CPS**)	Pyrimidine biosynthesis	*L. donovani*	Peritoneal murine MΦ BALB/c mice	↓ parasitemia	[82]
	Uracil phosphoribosyltransferase (**UPRT**)					
LdBPK_171470.1	Arabino-1, 4-lactone oxidase (**ALO**)	Ascorbate biosynthesis	*L. donovani*	J774.A1 murine MΦ BALB/c mice	↑ NO levels ↑ Th1 response ↓ parasitemia Vaccination confers protection upon challenge with a virulent strain	[32,83]
LDHU3_28.1280 LmjF.28.0980	**P27** protein	Part of mitochondrial cytochrome c oxidase complex enrolled in ATP synthesis	*L. donovani* *L. major*	Murine peritoneal MΦ Monocyte-derived human THP-1 MΦ BALB/c mice Dogs	↑ NO levels ↑ pro- and anti-inflammatory cytokine ↑ specific IFN-γ production by CD4^+^ and CD8^+^ T cells ↓ parasitemia Vaccination confers protection upon challenge with a virulent strain Homologous and heterologous protection Long-lasting protection Absence of lesion ↓ delayed-type hypersensitivity (DTH) reaction	[84,85,86,87]
LINF_060014300 LmjF.06.0860	Dihydrofolate reductase/thymidylate synthase (**DHFR-TS**)	Pyrimidine biosynthesis	*L. major*	BALB/c mice Rhesus monkey	Unable to sustain infection in vivo Long-term ↓ parasitemia Protective immune response	[88,89,90]
LdBPK_060370.1	Glutamine Synthetase (**GS**)	Promotes glutamate-ammonia ligation during glutamine biosynthesis	*L. donovani*	Monocyte-derived human THP-1 MΦ; BALB/c mice	↓ infectivity ↓ parasitemia Increased sensitivity to miltefosine	[91]
LmjF.25.0010	Beta galactofuranosyl transferase (**LPG1**)	Lipophosphoglycan biosynthesis	*L. major*	Murine peritoneal MΦ	↓ parasitemia	[92]
LmjF.36.2380 LmjF.36.2390	Sterol 24-c-methyltransferase (**C-24 SMT** or **ERG6**)	Ergosterol biosynthesis	*L. major*	BALB/c mice	↓ parasitemia	[93]
LmxM.23.0110	Mannose-1-phosphate guanyltransferase (**GDP-MP**)	Essential for the formation of GDP-mannose	*L. mexicana*	BALB/c mice	↓ parasitemia Susceptible to complement Long-lasting protection	[94]
LmjF.33.0830	2,4-dienoyl-CoA reductase (**DECR**)	Oxidoreductase in fatty acid β-oxidation	*L. major*	Murine BMDM BALB/c mice	↓ parasitemia	[95]
LmjF.30.0120	Alkyldihydroxyacetonephosphate synthase (**ADS1**)	Ether lipid biosynthesis	*L. major*	Murine peritoneal MΦ	↓ parasitemia Susceptible to complement	[96]
LMJLV39_040017200 LdBPK_041170.1	Fructose-1-6-bisphosphatase (**FBP**)	Phosphoric ester hydrolase in gluconeogenesis	*L. major*	Murine BMDM BALB/c mice	↑ levels of NO ↑ increased IFN-γ/IL-10 ratio ↑ Th1 response ↓ parasitemia Vaccination confers protection upon challenge with a virulent strain	[97,98]
LdBPK_120105.1	Ornithine decarboxylase (**ODC**)	Ornithine decarboxylase in polyamine biosynthesis	*L. donovani*	BALB/c mice	↓ parasitemia	[99]
LdBPK_040570.1	Spermidine synthase (**SPDSYN**)	Converts putrescine to spermidine	*L. donovani*	BALB/c mice	↓ infectivity	[100]
LdBPK_161100.1	Ubiquitin fold modifier 1 (**Ufm1**)	Protein ufmylation	*L. donovani*	Human elutriated MΦ	↓ intracellular amastigote survival in vitro and ex vivo	[101]
LinJ.28.3040	Heat shock protein 70 (**HSP70**)	Chaperone for the proper folding of the newly synthesized proteins	*L. infantum*	Human histiocytic lymphoma U937 cell BALB/c BALB SCID, (CB17^scid^), C57BL/6, Syrian golden hamsters	↑ levels of NO ↑ Th1 response ↑ specific IFN-γ production by CD4^+^ and CD8^+^ T cells ↓ decrease IL10 Production of IgG2a ↓ parasitemia ↓ disease evolution Vaccination confers protection upon challenge with a virulent strain Homologous and heterologous protection	[31,56,102,103,104]
LdBPK_354830.1	Peptidyl-prolyl cis-trans isomerase (cyclophilin-40—**Cyp40**)	Peptidyl-prolyl isomerase activity for proper protein folding	*L. donovani*	Murine BMDM C57BL/6 mice	Unable to sustain the infection in vitro	[105,106]
LINF_260007100	Silent information regulatory regulator 2 related protein 1 (**SIR2RP1**)	Histone deacetylation, leading to chromatin condensation and transcriptional silencing	*L. infantum*	Monocyte-derived human THP-1 MΦ;	Unable to sustain the infection in vivo ↑ levels of NO ↑ increased IFN-γ/IL-10 ratio Generation of specific anti-*Leish* IgG Ab subclasses Vaccination confers protection upon challenge with a virulent strain	[107]
LdBPK_231450.1	Sir2-family protein-like protein (**SIR2**) *	NAD-dependent protein deacetylase	*L. donovani*	J774A.1 murine MΦ	↓ infectivity	[77,108]
LmjF.11.0550	Nucleobase transport (**NT4**)	Purine uptake	*L. major*	BALB/c BMDM	Impaired survival of intracellular amastigotes	[109]
LdBPK_060310.1 LdBPK_100360.1 LdBPK_100370.1 LdBPK_100380.1 LdBPK_100390.1 LdBPK_100400.1 LdBPK_100410.1 LdBPK_100420.1 LdBPK_101450.1 LdBPK_190870.1	Folate/biopterin transporter (**BT1**)	Biopterin transporter for folate biosynthesis	*L. donovani*	BALB/c mice	↑ increased IFN-γ ↓ parasitemia Vaccination confers protection upon challenge with a virulent strain	[37]
LdBPK_044290.1	Golgi GDP-mannose transporter (**LPG2**)	GDP-mannose transporter LPG2 required for phosphoglycan synthesis	*L. donovani*	BALB/c BMDM	Unable to sustain the infection in vivo ↓ parasitemia Highly susceptible to complement-mediated lysis	[110]
LmjF06.0080 LmjF06.0090	ATP-binding cassette protein subfamily G, members 1 and 2 (**ABCG1** and **ABCG2**)	ATPase-coupled transmembrane transporter associated with drug resistance	*L. major*	Murine peritoneal MΦ; BALB/c mice	↓ infectivity ↓ parasitemia	[111]
LAMA_000100700	Mitochondrial carrier protein (**MIT1**) *	Mitochondrial iron importer	*L. amazonensis*	C57BL/6 and BALB/c mice	↓ parasitemia Absence of lesions	[112]
LmxM.36.6300 LmxM.36.6290 LmxM.36.6280	Glucose transporter LmGT (**GT1**, **GT2**, **GT3**)	Nutrient uptake	*L. mexicana*	Murine peritoneal MΦ; BALB/c mice	↓ infectivity ↓ parasitemia Smaller lesions	[113,114]
LmxM.36.5850 LinJ.36.6110	Kharon (**KH**)	Mediates transit of GT1 from the flagellar pocket into the flagellar membrane via interaction with the proximal portion of the flagellar axoneme	*L. mexicana* *L. infantum*	Murine peritoneal MΦ; Monocyte-derived human THP-1 MΦ; BALB/c mice C57BL/6 mice-γ^−/−^	↓ parasitemia Unable to sustain the infection in vitro ↑ IFN-γ/IL-10 ratio ↑ IgG; ↑ IL-17	[57,58,59]
LmjF.10.0460 LmjF.10.0465 LmjF.10.0470 LmjF.10.0480	Leishmaniolysin **GP63**	Surface metalloprotease involved in host–parasite interactions	*L. major*	BALB/c mice	Small lesions Highly susceptible to complement-mediated lysis	[115]
LdBPK_342160.1 LmjF34.2390 LbrM2903_20_2320 LmxM33.2390	Centrin 1 (**Cen1**)	Calcium-binding cytoskeletal protein essential for centrosome duplication or segregation	*L. donovani* *L. major* *L. braziliensis L. mexicana*	Murine peritoneal MΦ; BMDM and BMDC from BALB/c mice; C57BL/6 mice; Syrian golden hamsters; Dogs	Unable to sustain the infection in vivo ↑ NO levels ↑ Th1 response ↑ secretion IFN-y, IL-2, TNF-α and IL-12/IL-23p40 ↓ anti-inflammatory cytokines like IL-10, IL-21 ↑ Th17 ↑ IgG2a Absent or small lesions ↓ parasitemia Vaccination confers protection upon challenge with a virulent strain Homologous and heterologous protection Long-lasting protection	[29,33,34,35,36,53,54,116,117,118,119,120,121,122]
LmxM.29.0350	DEATH kinesin (**KIN29**)	Motor protein playing important roles in cell division, intracellular organization and flagellum formation and maintenance	*L. mexicana*	BALB/c mice	Absence of lesion or disease	[123]
LmxM.27.1620	Eukaryotic translation initiation factor 4E-1 (**EIF4E1**)	Cap-binding translation initiation factor	*L. mexicana*	RAW 264.7 MΦ	↓ infectivity	[72]
LAMA_000267300	META domain/Domain of unknown function (**DUF1935**)	Involved in metacyclogenesis	*L. amazonensis*	Murine peritoneal MΦ; BALB/c mice	↓ parasitemia	[124]
LmxM.19.0680	Flagellum attachment zone protein 7 (**FAZ7**)	Flagellum attachment zone connecting the base of the flagellum to one side of the flagellar pocket	*L. mexicana*	Monocyte-derived human THP-1 MΦ; BALB/c mice	↓ proliferation ↓ pathogenicity	[125]
LdBPK_061160	Noncatalytic component of GPI-mannosyltransferase (**GPI-MTI**) complex	GPI anchor biosynthetic process	*L. donovani*	BALB/c mice	Unable to sustain the infection in vivo Vaccination confers protection upon challenge with a virulent strain	[73]
LdBPK_290860.1	Cysteine peptidase C (**CPC**)	Cysteine peptidase proteolysis	*L. donovani*	U937 MΦ	↓ infectivity ↓ parasitemia	[126]
LdBPK_180150.1	Serine/threonine protein phosphatase type 5 (**PP5**)	Regulates HSP83 phosphorylation	*L. donovani*	RAW 264.7 MΦ; BALB/c mice	Absence of lesion	[127]
LmjF.36.5370	Leishmania protein-tyrosine phosphatase (**LPTP1**)	Protein tyrosine phosphatase	*L. donovani*	BALB/c mice	↓ parasitemia	[128]
LmjF.35.4180	Bardet-biedl syndrome 1 protein-like (**BBS1**)	Yet to be described, but in humans is required for specific trafficking events to and from the primary cilium	*L. major*	Murine peritoneal MΦ from BALB/c mice; Monocyte-derived human THP-1 MΦ;	↓ infectivity ↓ parasitemia Absence of lesion	[129]
LdBPK_320410.1	ATP-dependent RNA helicase (**HEL67**)	Essential for RNA metabolism, amastigote differentiation, and infectivity	*L. donovani*	Syrian golden hamsters	↑ NO levels Antigen-specific delayed-type hypersensitivity (DTH) Long-lasting protection	[130]
LmjF34.3940	Target of rapamycin kinase 3 (**TOR3**)	Regulators of cell growth, proliferation, and structure in eukaryotes	*L. major*	BALB/c mice	↓ parasitemia Absence of lesion	[131]
LINF_210009500 LmjF.21.0410	PIWI-like protein (**PWI1**)	Mitochondrial atypical argonaute-like protein that behaves as a programmed cell death sensor	*L. infantum* *L. major*	BALB/c mice	↓ parasitemia Delayed disease pathology	[132]
LdBPK_343830.1	Peptidase family C78—Ubiquitin Fold Modifier Protein Ufm1 Processing Peptidase (**UFSP**)	Abolishes Ufm1 processing	*L. donovani*	Human elutriated MΦ; BALB/c mice	↓ parasitemia	[133]
LmjF.08.0450	Signal peptidase type I (**SPase I**) *	Responsible for removing the signal peptide from secretory pre-proteins and releasing mature proteins to cellular or extra-cellular space	*L. major*	Murine BMDM BALB/c mice	↓ parasitemia Absence of lesion	[134]
*Targets whose overexpression leads to attenuation profile*						
LmxM.24.1770	Inhibitor of cysteine peptidase (**ICP**)	Regulation of proteolysis by inhibiting cysteine-endopeptidase	*L. mexicana*	BALB/c, CH3 and C57BL/6 mice	↑ Th1 response Significant control of growth or wound healing	[135]
LmjF.23.1040 LmjF.23.1082 LmjF.23.1070 LmjF.23.1080 LmjF.23.1086	Locus harboring: Hydrophilic acylated surface proteins (**HASPA1**, **HASPA2**, **HASPB**) Small hydrophilic endoplasmic reticulum-associated proteins (**SHERP1**, **SHERP 2**)	Overexpressing parasites showed decreased expression of glycoprotein GP63	*L. major*	BALB/c mice	↓ parasitemia Susceptible to complement	[136]

* Single knockout mutants where complete gene disruption was not possible. BMDM: bone-marrow-derived macrophage; BMDC: bone-marrow-derived dendritic cells; MΦ: macrophage; DTH: delayed-type hypersensitivity; NO: nitric oxide; up and down arrow symbols means increase and decrease, respectively.

## Data Availability

The data presented in this study are available in the Appendix A, in the main text, and in Table 1.

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
