# Peer review of "Next-Generation Leishmanization: Revisiting Molecular Targets for Selecting Genetically Engineered Live-Attenuated Leishmania"

_microorganisms, 2023, doi:10.3390/microorganisms11041043_

Round 1

Reviewer 1 Report

Overall, this review describes the development of genetically engineered live attenuated parasites, their immune response in animal models and pros and cons of live attenuated parasites. It is well researched article and has covered the literature extensively.

However, this review will benefit from describing how such vaccines can be used as alternative to leishmanization to further the development of Leishmania vaccines. In addition, the authors should further elaborate on the pros and cons of using such vaccines as alternative for leishmanization.

Even though the authors have made some references describing the immune response of genetically engineered live attenuated parasites as vaccines it does not elaborate on how such vaccine will afford protection against infection and disease.

Reviewer 2 Report

The manuscript (review) deals with a pretty interesting topic: the possible recovery of an ancient practice (leishmanization) to prevent leishmaniasis. As the authors comment, the advances in genetic manipulation of the protozoan Leishmania are being used to generate attenuated parasites lacking pathogenicity but able to induce a protective immune response that would protect vaccinated individuals from infection with virulent strains. However, the manuscript suffers from poor elaboration, and the authors should consider rewriting many parts of the manuscript. In the following points, some suggestions are included.

- Lines 17-18. Revise the sentence: discussing the their function

- Lines 50-53. Please, revise the sentence; its meaning is confusing.

- Line 72. Consider to use the verbal conditional tense, ie, vaccination would be the main

- Line 80. Check the meaning of 'fractionated Leishmania antigen'

- Lines 81-83. Please rephrase the sentence. Also, check whether the quoted reference is appropriate

- Line 90. Revise: consisted in inoculate infective.

- Lines 147-149. Please, rephrase the sentence. It is confusing

- Lines 160-161. Please, rewrite this part of the sentence: sequences that flank the targeted (Dihydrofolate reductase-thymidylate synthase, DHFR-TS).

- Line 162. Revise: replace homologous genes with foreign DNA.

- Line 188. Please, revise the link. It does not work.

- Lines 201-205. Please, revise the sentence. It is confusing.

- Lines 238-239. The authors should consider explaining the studies (unpublished) in more detail or deleting the sentence. In its present form, the sentence has little value.

- Lines 240-242. The sentence is confusing.

- Lines 349-350. Please, add a quotation or modulate the sentence.

- Line 357. Please, rewrite this part of the sentence: was recently published worth the reading.

- Line 380. Revise if it has sense: FBP can also be converted into glucose-6-phosphate

- Line 410. Please, elaborate further the sentence: Its deletion results in injury increase in mice.

- Line 430. Please revise: are mostly performed murine models.

- Lines 451-453. Please, revise the sentence; it is confusing

- Lines 453-455. Please, rephrase the sentence; it is confusing.

- Lines 457-458. Please, elaborate further on the sentence and include appropriate references.

- Line 465. What do you mean with 'as a precursor to human clinical trials'?

- Lines 468-470. Please, elaborate further the sentence.

- Line 494. Check this part of the sentence: needle are commonly used.

- Lines 494-497. The sentence is confusing; please, rewrite.

- Lines 504-507. Please, rephrase the sentence.

- Line 526. Please, rewrite; it is confusing.

- Lines 545-547. Please, rewrite the sentence; it is confusing.

- Line 547. What animals are you referring to with 'experimental animal model of CL'

- Lines 552-553, arguments in support of this sentence should be added by the authors.

- Lines 555-557. Please, rewrite and elaborate further this sentence.

Reviewer 3 Report

The manuscript from Moreira et al. is a deep review on the utility of the use of live-attenuated Leishmania parasites as vaccines. It also briefly describes and evaluates different techniques for gene manipulation in these parasites, including the most recent ones. The article describes 54 gene targets whose manipulation generates live attenuated parasites. It also briefly reviews the different animal models that have been used for vaccine evaluation and their advantages and drawbacks. Finally, the authors summarize the requisites that a human vaccine based on live-attenuated Leishmania must fulfil.

This thorough review of the topic can be considered of great interest to potential readers. The article is well organised. After checking the exhaustive list of references, I have not found any improper citation.  

Only very moderate changes in the text should be performed:

Line 296: have been reported instead of have reported.

Line 304: need

Lines 391-394: difficult to understand. Rewrite

Line 469: had

Line 469: was not protected instead of fails to protect.

Line 494: no comma after Although

Line 497. Rewrite

Line 537: no comma after mutants

Line 51: contributes

Reviewer 4 Report

In this review the authors review the molecular targets that can be considered for developing genetically engineered live-attenuated Leishmania vaccines, based on preclinical results of some live-attenuated vaccine candidates to date. The scope is appropriate and, despite there is not new information or discussion compared to other similar published reviews, the focus of this work is interesting.

The provided background is appropriate and informative enough, and the main text is well structured and sufficiently discussed. The effort to recapitulate the existing information and determine the best vaccines among 54 candidates using rational criteria is remarkable. However, there are differences in the amount of information used for each candidate (some of them have been extensively reviewed while some information from other candidates is missing) that can introduce some bias in this analysis. The most important aspects to be considered in the development of LAVs have been mentioned. I would suggest the inclusion of problems in the distribution of LAVs related to the maintenance of low temperatures.

Minor points

Line 443: I would suggest the inclusion of RT-PCR analysis to study immune responses in hamster model.

Line 132: Leishmania should be italicized

Line 240: describes should be described

Table 1: IGg2a in LinJ.28.3040 target should be IgG2a

Line 457: Immunosuppressed should be immunosuppressed

Line 479: a spacebar is needed before This

Line 483: I would suggest to include limiting dilution assay to detect viable parasites

Line: 542:  in vivo should be italicized

Round 2

Reviewer 2 Report

After reading the revised manuscript and the authors’ responses to my previous comments, I feel that the authors have addressed my suggestions sufficiently and improved the manuscript. Thus, I recommend accepting the manuscript for publication at this stage.